

# Nursing home residents' ADL status, institution-dwelling and association with outdoor activity: a cross-sectional study

Anne Marie Sandvoll[1], Ellen Karine Grov[2] and Morten Simonsen[3]

[1] Department of Health and Caring Sciences, Western Norway University of Applied Sciences, Førde, Norway
[2] Faculty of Health Sciences, Department of Nursing and Health Promotion, Oslo Metropolitan University, Oslo, Norway
[3] Department of Environmental Sciences, Western Norway University of Applied Sciences, Sogndal, Norway

## ABSTRACT

**Introduction**. The Norwegian regulations for nursing homes consider access to meaningful activities to be an indicator for the quality of nursing homes. Activities of daily living (ADL) provide important basic self-care skills for nursing home residents. Due to the physical changes caused by ageing and comorbidities, nursing home residents may experience functional decline over time, which may affect their ability to perform meaningful ADL, such as outdoor activity, which is considered a valuable and meaningful activity in Norwegian culture. This study aimed to investigate the association between ADL status, institution-dwelling and outdoor activity among nursing home residents.

**Methods**. This cross-sectional study included 784 residents aged >67 years living in 21 nursing homes in 15 Norwegian municipalities between November 2016 and May 2018. The Barthel Index was used to assess the nursing home residents' ADL status. Other variables collected were age, gender, body weight and height, visits per month, institution, ward, and participation in weekly outdoor activities. Descriptive statistics were used to provide an overview of the residents' characteristics. A Poisson regression model was used to test the association between the outdoor activity level as the dependent variable and ADL score, institution, and other control variables as independent variables.

**Results**. More than half (57%) of the nursing home residents in this sample did not go outdoors. More than 50% of the residents had an ADL score <10, which indicates low performance status. Further, we found that residents' ADL status, institution, ward, and number of visits had an impact on how often the residents went outdoors.

**Discussion**. The nursing home residents in this study rarely went outdoors, which is interesting because Norwegians appreciate this activity. Differences in the number of visits might explain why some residents went outdoors more often than other residents did. Our findings also highlight that the institutions impact the outdoor activity. How the institutions are organized and how important this activity is considered to be in the institutions determine how often the activity is performed.

**Conclusion**. The low frequency of the outdoor activities might be explained by a low ADL score. More than 50% of the residents had an ADL score <10, which indicates low performance status. Despite regulations for nursing home quality in Norway, this result suggests that organizational differences matter, which is an important implication for further research, health policy and practice.

Corresponding author
Anne Marie Sandvoll,
annemsa@hvl.no

## INTRODUCTION

Norway is an example of the Nordic welfare model and its welfare state is characterized by public funding and service provision (*Esping-Andersen et al., 2002*). Norwegian nursing homes are publicly financed, and the municipalities are responsible for providing this service. Access to meaningful activities is a reference for the quality of nursing homes as highlighted in Norwegian regulations for nursing homes (*Ministry of Health and Care Services of Norway, 2003*). This regulation, with its specific recommendations, can be used as an indicator to assess the quality of care in nursing homes (*Kirkevold & Engedal, 2006*). The regulations require the municipalities to ensure that each resident is offered varied and customised activities in line with other fundamentals of care (*Ministry of Health and Care Services of Norway, 2003*).

The availability of activities for nursing home residents may contribute to their well-being and dignity (*Björk et al., 2017*; *Lampinen et al., 2006b*; *Slettebø et al., 2016*). By contrast, according to *Nåden et al. (2013)*, the lack of participation in activities in nursing homes may be explained by the residents' physical impairments, e.g., some residents need to use wheelchairs. Up to 80% of nursing home residents experience cognitive impairment (*Selbaek, Kirkevold & Engedal, 2007*), which may also limit their ability to participate in activities such as playing cards, bingo and reading groups (*Strøm, Ytrehus & Grov, 2016*).

The outdoor lifestyle traditionally holds a prominent position in Norwegian culture (*Gurholt & Broch, 2019*) and is considered as a valuable and meaningful activity. Unfortunately, recent inspections undertaken by the authorities in nursing homes in Norway show a lack of activity offerings (*Helsetilsynet, 2018a*; *Helsetilsynet, 2018b*; *Helsetilsynet, 2018c*). The limited activity options indicate that the government's current policy and new regulations to increase the level of activities in Norwegian nursing homes have not yet succeeded (*Helsetilsynet, 2018a*; *Kjøs & Havig, 2016*; *Sandvoll, Hjertenes & Board, 2020*; *Sandvoll, Kristoffersen & Hauge, 2012*).

Despite the new regulations, changing nursing home practices is difficult (*Sandvoll, Kristoffersen & Hauge, 2012*). According to *Palacios-Ceña et al. (2015)*, nursing homes should strive to develop meaningful activities for residents to occupy their time and to provide residents with a meaningful sense of purpose. However, low levels of activities of daily living (ADL) among the residents can affect their ability to participate in activities (*Bürge, von Gunten & Berchtold, 2012*). ADL are an important basic self-care skill for the general population as well as for nursing home residents. Because of physical changes associated with ageing and comorbidities, nursing home residents may experience functional decline over time (*Drageset, Eide & Ranhoff, 2011*; *Liu et al., 2015*). Reduced ADL status may impair the ability to perform activities and can impact quality of life, social contact and loneliness (*Liu et al., 2014b*).

Physical activity, rehabilitation or exercise may improve independence and prevent the decline in ADL in elderly residents in long-term care facilities (*Bürge, von Gunten & Berchtold, 2012*; *Crocker et al., 2013*; *Liu et al., 2014b*). It is unclear which interventions are most appropriate for slowing the decline in ADL (*Crocker et al., 2013*), but it has been suggested that health professionals should promote physical activities with the aim of improving ADL performance among older adults (*Bürge, von Gunten & Berchtold, 2012*). The loss of ADL independence is the strongest predictor of the need for institutionalization of the elderly (*Gaugler et al., 2007*).

Several factors might influence nursing home residents' ADL status. Previous research has investigated the importance of ADL related to different aspects, such as loneliness, less participation in activities and depression. *Drageset (2004)* has shown that dependence in ADL status is associated with a high level of social loneliness. *Drageset, Eide & Ranhoff (2011)* later showed that greater dependence in ADL was associated with more symptoms of depression. Poor balance, incontinence, impaired cognition, low body mass index (BMI), impaired vision, no daily contact with proxies, impaired hearing and the presence of depression were significant risk factors for nursing home residents who experienced a decline in ADL status (*Bürge, von Gunten & Berchtold, 2012*).

Few studies have focused on the relationships between ADL status and participation in different activities among nursing home residents. One study investigated physical and social aspects of residents' mobility level and reported that nursing home residents dependent on a wheelchair or elevator during care were less involved in physical and social activities compared with more-mobile residents (*Kjøs & Havig, 2016*). This study suggests that reduced mobility might influence participation in different activities offered in the nursing homes. The need for activities and engagement in nursing home residents is well known (*Björk et al., 2017*; *Kjøs & Havig, 2016*; *Lampinen et al., 2006a*; *Palacios-Ceña et al., 2015*; *Theurer et al., 2015*). More research is needed on residents' ADL status and its relationship with participation in different activities, such as going outdoors.

Despite Norwegian regulations (*Ministry of Health and Care Services of Norway, 2003*), the frequency and content of activities are very much up to each nursing home. Previous studies have shown differences between privately owned and government-owned facilities (*Liu, Feng & Mor, 2014a*). Furthermore, previous studies have shown variations in practice regarding activities in Norwegian nursing homes (*Isaksen, Agotnes & Fagertun, 2018*). To the best of our knowledge, however, little is known on differences between institutions regarding their outdoor activities.

The aim of this study was to investigate the association between nursing home residents ADL status, institution-dwelling and outdoor activity. The following research question was formulated: *To what extent are nursing home residents' ADL status and the institution they live in associated with outdoor activity?*

## METHODS

A cross-sectional design was used.

### Setting

The data were collected by first-year nursing students during their placement in nursing homes between November 2016 and May 2018. The placement was either during the autumn semester, i.e., 8 weeks from the middle of October until the middle of December, or 8 weeks during the spring semester, from the middle of April until the middle of June. The data were collected during the daytime by means of a study manual, which the students had been presented in lectures at the university. For standardized instruments and questionnaires, we used the connected manual, procedure or protocol. The process of data collection was supervised by the university teacher and the nurses working at the different nursing homes. Each patient was registered once.

Our responses were collected from 21 different nursing homes. These institutions differ because they have different combinations of ward types and may have different attitudes towards outdoor activity. All nursing homes in this study except for one are financed and operated by the municipality. The single private nursing home is not run by a commercial actor, but by the parish associated with the Bergen Cathedral in Bergen, the second largest city and municipality in Norway. Nursing homes all share the same national financing system.

### Participants

The study included 784 residents aged >67 years living in 21 nursing homes in 15 Norwegian municipalities. The inclusion criteria were all residents aged >67 years living in the selected nursing homes, while the exclusion criteria were residents receiving palliative care, related to ethical considerations, to protect them from harm related to the completion of questionnaires in their presence. In addition, residents in the palliative phase may be unable to take part in the outdoor activities described in this paper. Five of the nursing homes were located in rural areas, while others were located in small villages. The nursing homes were not selected completely at random because the selection was partially determined by what nursing homes the nursing students attended during their practice period.

### Variables

We expected increasing levels of outdoor activities, e.g., making trips outside the nursing home, with increasing *ADL score* because ADL is a measurement of physical capability (higher scores mean better capabilities). We observed the residents by using the method described in the Barthel Index for measuring performance in ADL, as translated and revised by *Saltvedt et al. (2008)*. Each performance item is rated on this scale with a given number of points assigned to each level, related to how dependent or independent the resident is, with maximum of 20 points (20 = totally dependent). The Barthel Index is a standardized, validated and psychometric-tested instrument widely used in the context of elderly care (*Liu et al., 2015*; *Mahoney & Barthel, 1965*).

*Outdoor activity* is the dependent variable in our analysis. In this study, the residents either walked on their own or with assistance from staff or visitors. Some residents went

outdoors with a walker or in a wheelchair. Some of the residents had an electric wheelchair and went outside on their own. However, the purpose was still the same: outdoor activity. The level of this activity was measured and documented as the number of times the activity was performed during a week. Further, we introduce *nursing homes* as random effects to allow for the fact that not all types of nursing homes are included. These effects will tell us whether activity levels vary between institutions. We included a dummy variable for residing in a short-term/rehabilitation ward and one for residing in a dementia ward. Long-term ward residents are expected to be older, frailer and in need of more care; thus, we expected these residents to have the lowest levels of making trips outside the nursing home.

The *number of visits* (per week) is interpreted as a proxy for less social isolation (*Drageset, 2004*). We expected that more visits would lead to higher levels of outdoor activities. More visits may also mean that relatives engage in this activity, which increases the level of ADL. Further, we expected decreasing levels of activity with increasing *age* (*Feng et al., 2017*). The gender dummy variable was coded as 1 for men and 0 for women. We had no specific expectations for a gender effect on making trips outside the nursing home.

*BMI* is an indication of the general health condition. A low BMI indicates that residents are not eating enough (or that they fail to maintain their body weight). We expected that low BMI would be associated with fewer trips outside the nursing home.

All variables were registered in a form and documented in Excel version 16.16.19 (Microsoft, Redmond, WA, USA).

## Bias

There are some limitations in using this approach. The sample is not completely randomized since the nursing homes were not selected at random, neither were the residents in the nursing homes. The students may also understand the concept differently or they did not apply it consistently. However, a detailed protocol was provided to the students so that their observations were made consistently. For instance, what date format should be used, and age and length of stay should be integer numbers. We could not eliminate ambivalence in the data collection completely.

## Statistical methods

Descriptive statistics were used to give an overview of the demographic and clinical characteristics of the participants, including age, gender, BMI, ADL status, institution and the prevalence of residents' outdoor activities. We sorted the informants into different groups according to the quartiles from the distribution of ADL scores. We then analysed the levels of the outdoor activities between these groups. To further examine the association between ADL score and outdoor activity, we included age, BMI, gender, visits per month, type of ward and ADL score as well as institutions in a multivariate Poisson regression model. The data were analysed using the SAS GLIMMIX procedure with a Poisson log-link function. The two-sided significance level was set to 0.05.

We designed a model with outdoor activity as the dependent variable and ADL status as an explanatory variable affecting the level of this activity. In addition, we controlled for

several other explanatory variables that may have an influence on both activity level and ADL scores, thereby eliminating possible spurious factors. We also included institutions as an independent variable, assuming they are random effects, which allows the coefficients to vary between institutions.

We assume that institutions (nursing homes) represent several unmeasured characteristics that vary between them. These characteristics may be different service quality, different organizations, different informal routines established among staff, different efficiency in using resources, or different resident characteristics. These characteristics are not measured and probably cannot be measured. Institutions are clusters therefore we include them as random effects in the regression model to account for these variations. On the other hand, the impacts of different ward types are fixed effects since ward types have the same definition for all nursing homes and therefore do not measure any latent characteristics. The model is estimated as random intercept model, each institution has an individual- specific random effect in addition to the fixed effects of all other independent variables (*SAS Institute, 2019*). A mixed model with both fixed and random effects that is designed to capture variations between clusters is called a conditional model (*Muff, Held & Keller, 2016*).

The model allowed us to control for other regressors when assessing the effect of ADL score or institutions on outdoor activity. Thus, we could compare activity level between residents in the same ward and with the same age, gender, number of visits per month and BMI, but with different ADL scores in different institutions.

Clustering occurs when entities are distributed on several levels. When this is the case, error terms within a cluster will not be independently distributed of error terms in another cluster (*Trutschel et al., 2017*). In our design, this means that error terms between nursing homes will be biased if they are not accounted for in the regression model. We have already considered different ward types because the chance of a resident performing the activity may be affected by the ward type in which the resident lives. Nursing homes (institutions) and ward types are two cluster types; therefore, we should also consider differences between nursing homes in the regression model.

The Table A1 (Appendix A) shows the goodness-of-fit values for the regression model with trips outside the nursing home in the preceding week as the dependent variable. The dispersion criteria $\chi^2$/df has a value <2. Therefore, we assumed no overdispersion in the model.

Further, we computed the intraclass correlation (ICC1) from the estimated model. This was performed by using a function in R, since SAS does not provide this statistic (*Lüdecke, 2020a*). The correlation is calculated as the random effect variance divided by the sum of this variance plus the residual variance. The conditional ICC1 takes the fixed effects into consideration as well as the random effects (*Lüdecke, 2020b*). The estimated conditional ICC1 is 0,145 which means the cluster effects account for almost 15% of the variation in dependent variable making trips outdoors. The model estimated in R (function glmer) gives the same results as with the GLIMMIX procedure in SAS using the Laplace maximum likelihood method.

**Table 1  Descriptive statistics for dependent variable, trips outdoors last week.**

| Trips outdoors last week | |
|---|---|
| Min | 0 |
| 25th percentile | 0 |
| Median | 0 |
| 75th percentile | 1 |
| 95th percentile | 4 |
| Max | 14 |
| Number of observations | 784 |

Except for the ICC1, all statistical analyses were performed using SAS software (University Edition; SAS Institute, Cary, NC, USA).

### Ethics

The Regional Medical Ethics Committee (REK West), University of Bergen (2015/2030 REK WEST, University of Bergen) and the Norwegian Social Science Data Services (46303) approved the study, which was endorsed by all nursing homes. Voluntary, written informed consent was obtained from all participants. In situations where the resident was not able to give consent related to e.g., dementia or cognitive impairment, either the resident's relatives or the department manager gave consent.

## RESULTS

The sample ($n = 784$), included more women (69%) than men (31%), , which is consistent with the population distribution in this age group (*Statistisk Sentralbyrå, 2016*). Most residents in our sample (55%) resided in a long-term facility, 26% resided in a dementia ward and 19% resided in a short-term ward. The mean ADL score was 10.1. We distributed residents into groups according to their ADL score using the quartiles from the ADL distribution, which resulted in about the same number of residents in each group. Twenty-eight per cent of the residents had an ADL score of 0–6 points as measured by the Barthel Index, 24% had an ADL score of 7–10 points, 26% had an ADL score of 11–14 points and 23% had an ADL score >15 points.

Table 1 shows descriptive statistics for the dependent variable, trip outdoors last week. Table 2 shows descriptive statistics for the continuous independent variables while Table 3 shows descriptive statistics for categorical independent variables.

Nursing homes are used as cluster variables for estimating random effects. The range is from 5 to 123 registrations per nursing home. The average was 37 registrations while the standard deviation was 31, suggesting high variation in registrations. The nursing homes also vary considerably in size which explains some, but not all of this variation.

Table 4 shows the results of the model estimation with outdoor activities in the preceding week as the dependent variable. The table shows the fixed effects in the model. The random effects are available in Appendix B (Table A2). These are obtained by using a random intercept model, one for each nursing home, which implies that the fixed effects are

**Table 2 Descriptive statistics for numeric independent variables.** Descriptive statistics for the continuous independent variables.

| | Mean | Standard deviation | Number of observations |
|---|---|---|---|
| ADL score | 10.1 | 5.2 | 787 |
| Age | 86.3 | 7.2 | 786 |
| Visits per month | 8.9 | 8.8 | 787 |
| Body mass index | 24.2 | 5 | 785 |

**Table 3 Descriptive statistics for categorical variables ward type and gender used as independent variables.**

| | N | % of total |
|---|---|---|
| **Ward type** | | |
| **Short term/rehab** | 153 | 19.4 |
| Long term | 434 | 55.1 |
| Dementia | 200 | 25.4 |
| Total | 787 | 99.9 |
| **Gender** | | |
| **Female** | 543 | 69 |
| Male | 244 | 31 |
| Total | 787 | 100 |

**Table 4 Model estimates of outdoor activities in the preceding week: Poisson regression.** The results of the model estimation with outdoor activities in the preceding week as the dependent variable.

| Effect | Estimate | Relative risk | Standard error | df | t value | Pr > \|t\| |
|---|---|---|---|---|---|---|
| Intercept | 0.925 | | 0.590 | 20 | 1.57 | 0.133 |
| ADL | 0.052 | 1.054 | 0.008 | 756 | 6.72 | <.0001 |
| Gender (1 = Male) | 0.102 | 1.107 | 0.086 | 756 | 1.18 | 0.238 |
| Age | −0.024 | 0.976 | 0.006 | 756 | −4.18 | <.0001 |
| Visit Pr month | 0.030 | 1.031 | 0.004 | 756 | 7.68 | <.0001 |
| BMI | −0.005 | 0.995 | 0.008 | 756 | −0.60 | 0.548 |
| Dementia | 0.462 | 1.588 | 0.115 | 756 | 4.00 | <.0001 |
| Short-term rehabilitation ward | −0.309 | 0.734 | 0.135 | 756 | −2.29 | 0.023 |
| AIC | 1,973.25 | | | | | |
| Variance random effect | 0.1532 | | | | | |

assumed to be constant over all nursing homes. Long-term ward type is the reference case for ward types and its effect is measured by the model's general intercept.

The ADL score has a significant impact on the activity. An increase in the ADL score of 1 was expected to give an increase in the rate of activity level of 1.05. We show this effect by considering two residents, both women aged 85 years, living in a long-term ward, receiving 6 visits per month and having a BMI of 23.8 kg/m$^2$ (the last two numbers are median values). Both women live in institution number 1. Resident A had an ADL score

of 10, while resident B had an ADL score of 15. From our model, we expected resident A to take 0.43 trips outside the nursing home in the preceding week and resident B to take 0.56 trips. Accordingly, we expected that 16 days would be needed for resident A to take one trip outdoors and 13 days would be needed for resident B. Had the two residents lived in institution number 7, the expected number of trips would have been 1.3 and 1.7 trips outdoors, assuming values for age, number of visits, BMI and gender stay the same and ADL score is 10 and 15, respectively, as above. In other words, both residents A and B would have three times more outdoor activities if they had been living in institution 7 instead of 1. This result shows that institutions have an impact on activity level. This is confirmed by estimation of institutional random effects (Table A2) where eight institutions have significant effects, four of them are positive.

Table 4 also shows that age, visits per month and ward type had significant effects on the number of outdoor activities during the week. All effects were as expected: i.e., increasing age was associated with a lower activity level, whereas an increasing number of visits were associated with more trips outside the nursing home. The relative risk factor for age shows that a resident with 80 years is assumed to have 0.8 less trips outdoors previous week compared to a resident with 70 years, assuming all other variables are held constant and that they reside in the same institution. Also, with the same assumptions, a resident receiving 10 visits per month is expected to have 1.3 more trips outdoors compared to a resident receiving only one visit per month.

The effects of short-term wards were negative, indicating the residents in that ward type took significantly fewer trips outside the nursing home than did residents in the long-term ward. On the other hand, residents in dementia wards took significantly more trips outdoors than residents in long-term wards. Based on the assumptions outlined above, the risk factors show that a resident in the dementia ward had almost 1.6 more trips outdoors in the previous week compared to a resident in the long-term ward. A resident in the short-term ward had 0.7 less trips. Gender and BMI had no significant effect on the number of outdoor activities.

## DISCUSSION

Our findings show that institutions are important predictors for the level of trips made outdoors by residents. Age and BMI index had significant effects, both negative since increasing age and BMI led to fewer trips outdoors. Visits per month had a significant positive effect, more visits led to more trips outdoors. In addition, ADL-score had a significant effect on the activity, the lower the score, the lower the activity level.

Further, our findings show that 57% of the nursing home residents in this sample did not go outdoors. This is consistent with other studies showing that the activities offered in nursing homes are limited (*Kjøs & Havig, 2016*) and that the residents often are inactive (*Harper Ice, 2002*). Recent inspections of nursing homes undertaken by the Norwegian authorities confirm the lack of activity offerings (*Helsetilsynet, 2018a*; *Helsetilsynet, 2018b*; *Helsetilsynet, 2018c*).

The findings of our study might be explained by the residents' ADL score, which was low: i.e., 50% of the residents had an ADL score between 0 and 10. These low ADL scores

indicate that these residents had a low ability to go outdoors. This is consistent with national health policies in Norway, which emphasize that the frailest elderly should receive care in nursing homes. It is also in line with previous research that shows that the frailest residents might not be able to go outdoors because of their old age, fatigue, frailty or illness (*Nåden et al., 2013*).

However, *Björk et al. (2017)* performed a similar study in Sweden and reported that 60% of the nursing home residents had gone outdoors during the data collection period (November 2013–September 2014). The differences in going outside the nursing home in these similar studies from the Scandinavian health-care context are interesting. Weather and the need for appropriate clothing or equipment can impede the ability of residents to go outdoors. If *Björk et al. (2017)* collected data during the summer, it might explain some of these differences. Our data were collected either during autumn or spring. In Norway the temperature and weather conditions often are warmer and contain less rain during July and August, and the residents are more likely to go outdoors. This might explain why the residents in the Swedish study went outside more often (*Björk et al., 2017*). Further, our data were collected in the western part of Norway which has more rain compared to the eastern parts of Norway where most people live. In addition, these residents might not have proper clothing like raincoats, warm jackets, appropriate shoes or hats suitable for the different weather conditions. The British Broadcasting Corporation (*BBC, 2018*) has shown how the use of a rickshaw with a roof and cover may be an alternative for helping frail elderly people to perform outdoor activities despite their loss in ADL status. The concept of outdoor life, in particular hiking, has a prominent position in the Norwegian culture (*Gurholt & Broch, 2019*). In addition, most of the older population in Norway grew up after the last world war; therefore, many have received basic socialization in outdoor life and have maintained their association with outdoor activities throughout their lives (*Odden, 2008*).

Our findings highlight that institutions have an impact on how often residents go outdoors. These findings suggest that organizational differences impact outdoor activity. How the institutions are organized and the importance they give this activity obviously determine how often it is performed. These findings are in line with *Isaksen, Agotnes & Fagertun (2018)*, who found that only four of 17 nursing homes had activity plans for the wards. Further, they found variations in staff who had participated in training program regarding activities for the residents (*Isaksen, Agotnes & Fagertun, 2018*). Even if the service going outdoor is regulated by national regulations (*Ministry of Health and Care Services of Norway, 2003*), there is considerable room for adaption in each nursing home. The variation in service provision between nursing homes comes from different cultures, organizational practices and plainly the priority the service gets when set against other services the nursing homes are obliged to provide (*Nakrem, 2015*). To increase the level of activity, students should be given more information about the benefits of the activity for nursing home residents as well as the legal rights of this activity.

Physical activity is important for mental well-being among elderly people (*Lampinen et al., 2006b*). However, our findings show that increasing age was associated with lower activity levels, which is also in line with *Feng et al. (2017)*. This might imply a natural

change from being active to being less active and in need for assistance, which corresponds with the process of disengagement described by Cumming and Henry in 1961 (*Daatland & Solem, 2011*). When people get older, it is natural for them to gradually withdraw from their social roles and the activities they used to perform. This is in line with *Adams, Roberts & Cole (2011)*, who found that activity participation in late life changed from an active social life with creative activities to an increased participation in passive social and spiritual activities. Nursing homes must consider this and meet their residents' individual needs and interests. According to the Norwegian quality regulations, nursing home residents should be offered varied and customized activities (*Ministry of Health and Care Services of Norway, 2003*). Nursing homes need to facilitate activities that are suitable for each resident's ADL status and individual wishes. For example, it might be important for residents to have their own personal things near their own chair. A nearby table might contain personal important objects, such as magazines, books, newspapers or medicines (*Board & McCormack, 2018*). Nursing home residents who are no longer capable or do not want to go outside might appreciate a nice view (*Eijkelenboom et al., 2017*). Activities are a basic need and participation in activities might contribute to the well-being and dignity experienced by nursing home residents (*Björk et al., 2017*; *Lampinen et al., 2006b*; *Slettebø et al., 2016*). Such activities should be organized by the staff in close co-operation with relatives because they are familiar with the residents' needs (*Sandvoll, Kristoffersen & Hauge, 2012*).

Previous research shows that nursing home staff are committed to routines, such as helping residents with personal care, practical help, nutrition and toileting (*Harnett, 2010*; *Sandvoll, Kristoffersen & Hauge, 2012*), but do not always take a person-centred approach (*McCormack, 2016*) in terms of their activities. Nursing homes often lack the opportunity and time to offer activities for all residents and their staff recognize that some residents may spend time sitting alone even though staff members know that they might have preferred to join in activities (*Sandvoll et al., 2015*). Could the lack of staff explain our study results? Our findings show that visits per month and ward type had a significant effect on number of outdoor activities during the week. An increasing number of visits were associated with more trips outside the nursing home. This shows that the visits (from family or volunteers) have an impact on resident's level of activities regarding outdoor activity. In Norway, the government has addressed new ideas to solve the staff challenges and suggests that voluntary contributions by relatives and organizations should be included as a way of providing activities for nursing home residents (*Det kongelige kulturdepartement, 2018*; *Ministry of Health and Care Services of Norway, 2013*).

A reform to improve elderly care was introduced in a recent white paper from the Norwegian government. One of the main areas that need improvement in elderly care is activities for elderly people living in nursing homes and the white paper suggests that they should participate in one hour of activity every day (*Ministry of Health and Care Services of Norway, 2018*). To provide more activities for nursing home residents, particularly outdoor activities, nursing home staff should be given resources to organize individual, person-centred and customized activities for all residents and to co-ordinate voluntary contributions (e.g., from family members and elderly that want to participate in activities).

This is consistent with a recent study by *Skinner, Sogstad & Tingvold (2018)*, who found that the voluntary, unpaid contribution took place within cultural, social and other activities aimed at promoting mental stimulation and well-being. Furthermore, they suggested that the staff in government nursing homes should consider voluntary contributions when they plan the care of residents in long-term care (*Skinner, Sogstad & Tingvold, 2018*). To offer a variety of activities for nursing home residents, activities should be offered both inside and outside the nursing home. We also encourage the national authorities to specify in white papers that activities for Norwegian nursing home residents should take place both indoors and outdoors. For residents who are unable to go outdoors on their own, rickshaws might serve as an alternative way of enabling them to go outdoors. Our findings show that nursing home residents rarely engage in outdoor activities, even though the need for activities and engagement for nursing home residents is well known internationally (*Björk et al., 2017*; *Kjøs & Havig, 2016*; *Lampinen et al., 2006a*; *Palacios-Ceña et al., 2015*; *Theurer et al., 2015*). Therefore, a greater focus on activities for elderly nursing home residents should be increased and customized in line with each resident's individual needs and wishes. Finally, our results show that the institution that the residents live in has an important association with outdoor activity. This implies that organizational differences in nursing homes might have an impact on outdoor activity, which is an important implication for further research, health policy and practice.

## Strengths and weaknesses

The strength of this study is the systematic use of standardized, psychometric-tested instruments and measures (*Mahoney & Barthel, 1965*). One weakness is related to the nursing students' observations used to rate ADL. One obligation of research is not to harm participations; i.e., even though self-report is recommended as the gold standard for gathering data (*Polit & Beck, 2017*), self-report was considered to be inappropriate for assessing the ADL of these residents. The students' involvement in research might contribute to mutually strengthening research and education. The students used a predefined manual or standardized protocol to assess data, which is an advantage, particularly since the lecture was given immediately before clinical placement. The data collection was supervised by the university teacher and nurses working at the different nursing homes. This might, on the other side, be a bias in this study because the involvement might serve as a Hawthorne effect (*Polit & Beck, 2017*). The participants represent a convenient sample from clinical placements where the university has contracts educating students. In such a way, it might be limited possibilities for generalization of the results to all nursing homes.

## CONCLUSIONS

More than half (57%) of the participants in this study did not go outdoors during the preceding week. Their ADL status might explain this pattern because more than 50% of the residents had an ADL score <10, which indicates low performance status. The institutions that the residents live in have an impact on outdoor activity, which suggests that organizational differences matter. This is an important implication for further research, health policy and practice. Planning for nursing home residents' activities requires staff

competence in assessing the capacity and needs of all residents. Those residents with few family members or friends might benefit from visits from volunteers taking on an important function in collaboration with the nursing staff in managing different kind of activities, such as outdoor activities. Our findings show that residents rarely engage in outdoor activities, even though the need for activities and engagement for nursing home residents is well known. Therefore, a greater focus on activities for elderly nursing home residents should be increased and customized in line with each resident's individual needs and wishes.

## APPENDIX A

Table A1  Model statistics for Poisson regression model for outdoor activities in the preceding week as a dependent variable.

| Goodness-of-fit criteria | df | $\chi^2$ | $\chi^2$/df |
|---|---|---|---|
| Generalized chi-square | 774 | 1,216.3 | 1.57 |
| Number of observations | | 784 | |

## APPENDIX B

| Table A2 | Model estimates of outdoor activities in the preceding week—random effects. | | | | |
|---|---|---|---|---|---|
| Institution | Estimate | Standard Error | df | t value | Pr > \|t\| |
| InstId 1 | −0.430 | 0.1896 | 756 | −2.27 | 0.0236 |
| InstId 2 | 0.595 | 0.1793 | 756 | 3.32 | 0.0009 |
| InstId 3 | −0.247 | 0.1671 | 756 | −1.48 | 0.139 |
| InstId 4 | −0.031 | 0.2164 | 756 | −0.14 | 0.8872 |
| InstId 5 | −0.140 | 0.3083 | 756 | −0.45 | 0.6499 |
| InstId 6 | −0.453 | 0.2255 | 756 | −2.01 | 0.0451 |
| InstId 7 | 0.689 | 0.1519 | 756 | 4.54 | <.0001 |
| InstId 8 | −0.567 | 0.2713 | 756 | −2.09 | 0.0371 |
| InstId 9 | 0.099 | 0.2133 | 756 | 0.46 | 0.6441 |
| InstId 10 | −0.036 | 0.1487 | 756 | −0.24 | 0.8068 |
| InstId 11 | 0.630 | 0.3366 | 756 | 1.87 | 0.0615 |
| InstId 12 | 0.420 | 0.1742 | 756 | 2.41 | 0.0161 |
| InstId 13 | 0.155 | 0.2227 | 756 | 0.69 | 0.4881 |
| InstId 14 | −0.022 | 0.1521 | 756 | −0.14 | 0.8866 |
| InstId 15 | −0.284 | 0.2389 | 756 | −1.19 | 0.2343 |
| InstId 16 | 0.472 | 0.2612 | 756 | 1.81 | 0.0713 |
| InstId 17 | −0.518 | 0.2509 | 756 | −2.06 | 0.0395 |
| InstId 18 | 0.146 | 0.243 | 756 | 0.6 | 0.5478 |
| InstId 19 | −0.372 | 0.3453 | 756 | −1.08 | 0.2821 |
| InstId 20 | 0.021 | 0.3065 | 756 | 0.07 | 0.9453 |
| InstId 21 | 0.191 | 0.2434 | 756 | 0.79 | 0.4322 |

### Funding
This work was supported by Western University of Applied Sciences, Norway. The funders had no role in study design, data collection and analysis, decision to publish, or preparation of the manuscript.

### Grant Disclosures
The following grant information was disclosed by the authors:
Western University of Applied Sciences, Norway.

### Competing Interests
The authors declare there are no competing interests.

### Author Contributions
- Anne Marie Sandvoll conceived and designed the experiments, performed the experiments, authored or reviewed drafts of the paper, and approved the final draft.

- Ellen Karine Grov conceived and designed the experiments, authored or reviewed drafts of the paper, and approved the final draft.
- Morten Simonsen analyzed the data, prepared figures and/or tables, authored or reviewed drafts of the paper, and approved the final draft.

## Human Ethics

The following information was supplied relating to ethical approvals (i.e., approving body and any reference numbers):

The Regional Medical Ethics Committee REK West, University of Bergen (2015/2030 REK WEST, University of Bergen) and the Norwegian Social Science Data Services (46303) approved the study, which was endorsed by all nursing homes.

## Data Availability

The raw data are available in a Supplemental File.

## Supplemental Information

Supplemental information for this article can be found online at http://dx.doi.org/10.7717/peerj.10202#supplemental-information.

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
