# Peer review of "Nursing home residents' ADL status, institution-dwelling and association with outdoor activity: a cross-sectional study"

_PeerJ, doi:10.7717/peerj.10202_

## Round 0.1 · original submission · Major Revisions

We have received now two reviews, which indicate important revisions necessary for the paper to be considered for publication. In particular reviewers pointed out the organization of the manuscript, the reporting of the methodology and the language. Frankly, the paper still requires a mayor, mayor revision. You might consider to get professional editing and some methodological support.

·

Basic reporting

The structure of the article needs to be improved. Especially the method section is confusing to the reader, as details on measurement and data collection are given quite late in the manuscript, but which is needed earlier to understand everything that follows properly. Details on the statistics are given in the "Design" section that need to be placed in the "Analysis" section. These are just examples. I commented the structure in the pdf to make clear where improvement is needed.

In the results the authors come up with a result that was not part of the research questions and theoretical assumptions: the influence of the institution. If this result is presented it needs to be included in the research question, theoretically reasoned why the authors included this variable and discussed properly regarding the impact that this result has. Please also see my comments in the pdf.

The tables and ficures need to be revised (the number of tables shouls be reduced, captions added). Figure 1 need axis labels and a caption.

Experimental design

The meaningfulness of the original research question may be a bit questionable, as the investigated association is quite self-evident. This is also a reason why I recommend to include the influence the instituion has as a research question, as this is not so obvious and also not often investigated.

Validity of the findings

The discussion and the conclusion need to be broadened with respect to the finding that the institution has an effect on the dependent variable.

Reviewer 2 ·

Basic reporting

Some sentence structure and American English issues in sentence structure remain. Literature sources are provided except where noted. The article is more cohesive with this review, however, major flaws remain.

The paper is unorganized. Adding a full literature review would be helpful.

A description of the variables should be consistent and located in the methods section

Content provided under the methods heading is unclear

Experimental design

The design of the study is stated as cross sectional with randomized selection of homes. No description of how the randomization was conducted is provided. As these homes are locations where nursing students perform clinical, the reader feels the randomization may have been somewhat biased. A full description of how the randomization was done should be included. For example, were all homes in a specifed area able to participate? You were required to obtain consent from all resients, however, a cross sectional study implies all residents in every home (except those who were exclused due to exclusion criteria) participate. Not knowing how many declined is a major flaw in this study design.

You have one variable ADL as measured by the Barthel Index is valid and reliable (although those validity and reliability statistics were not provided). No other variables are. Your description of the variables is unclear and presented in two locations which is confusing. Your description of the variable "visits" "going outside" and "activity" are believed to be the same variable, however, it is unclear.

The analysis is described as Poisson model or a multivariate model, then stated that you chose to use a negative binomial regression model instead. You should consistently describe what method of analysis you use. In addition, you have different homes with different ward types and different Norwegian municipalities. Each home is nested within each Norwegian municipalities. You need to review your analysis.

Validity of the findings

Parts of the discussion are overstated, stated incorrectly, or have English language wording issues that may help to confuse the meaning. See comments in paper.

Suggestions are provided to strengthen the strengths and limitations of this paper. Expand on strengths, there are more than one strrength. Weakness can be more clear and could include:

This weakness needs to be addressed as potential research bias. Measurement bias arises from a potential error the data collection and the process of measuring.You are trying to say that the students might not have assessed residents accurately. There are ways to address this in your data collection, but you did not so address as possible research bias.
How were the activities residents participated in counted. Just going outdoors is difficult to count unless you have someone outside to track 24/7 for the entire period, this is also a measurement bias

Did anyone decide not to participate or is it expected in this culture to participate in research activities. This may be a bias as well. Your homes were randomly selected but were the residents? Inclusive bias occurs when samples are selected for convenience.

Participants might be biased as well. Any self-reported data could be provided in a way the participant believed the students wanted to hear. response bias is a type of bias where the subject consciously, or subconsciously, gives response that they think that the interviewer wants to hear.

Additional comments

Thank you for the opportunity to review this paper. The authors have conducted a major revision of this paper and as a result, a new manuscript has been submitted. The authors would benefit from some guidance on research methodology and English language prior to the next submission. Please see comments above and in the paper itself.

Annotated reviews are not available for download in order to protect the identity of reviewers who chose to remain anonymous.

---

## Round 0.2 · Major Revisions

CoVID-related work blocked any capacity for a while, and I am sorry for the delay in handling the paper.

Overall the paper has improved substantially. However, it still requires more work to be considered for publication. The two main points are the presentation of the results (tables, figures) and the organization of the text.

Major comments:

1) The method section is disorganized. Please use a structure like provided in the STROBE guideline. (https://www.equator-network.org/wp-content/uploads/2015/10/STROBE_checklist_v4_cross-sectional.pdf)
2) Please provide a clear account of the variables used in the analysis. Provide them in one section. I am still not sure how outdoor activities were measures. Please keep description of data collection, analysis and your suspected theoretical considerations separate. For instance line 129 – the variable section is not for described suspected associations between variables.
3) There is too much redundancy in the text. For instance, Lines 129-151 and 213-227. Please organize the text accordingly.
4) The descriptive reporting is not yet optimal. Please use one descriptive table, which contains all variables used in the model. For numeric variables with means and sd, for categories n (%). For the outdoor activities I would suggest using the median and min, max values. This should also replace figure 1.
5) Methods: Please provide the intraclass correlation based on the unconditional random effect model (ICC1). This will be above 0.05 and will strengthen your argument.
6) Methods: There is a high number of zeros. Did you consider a zero-inflated Poisson regression model?
7) Discussion: Please make sure you compare apples with apples, is the sampling and the instruments between your study and Björk 2017 comparable? This is a lengthy argument about the weather, and it is outside the data you collected. All in all, I am not convinced this is driving the observed data. Please assess again whether this is really comparable.

Minor comments:
Line 119 remove reference. It is to generic to be useful.
Line 240 “spurious causal factors” – this does not exist either it is a spurious association or a causal factor, but not both…
Line 265-274 Remove this section. I don’t think that this argument holds, since you do not have a random sample.
Line 309-312: This belongs if at all in the methods section, it’s pretty much redundant and I would suggest removing table 3.
Line 315-321: Move this to the methods section, this is part of the model specification.
Table 4: Please limit to three digits after the comma, do it consistently. Please no vertical lines, only one horizontal line under the header… (applies for table 1 too)
Line 283-285: Remove sentence

·

Basic reporting

x

Experimental design

The section „Limitations“ that is now placed in the Method section of the text should rather be placed in the “Strength and weaknesses” section.

Validity of the findings

x

---

## Round 0.3 · Minor Revisions

The paper has improved, yet authors did not adequately respond to some of the remarks. Therefore, please be careful in your response to the comments.

- Please provide the ICC1. You are right that the ICC1 is used for reliability assessment – but it is also a key indicator for clustering in multilevel models (1-4). This is very important in the context of your analysis as nursing home is a relevant factor.
- Please organize statistics section as descriptive and inferential analysis. Specify all variable included in the model as part of the description of the model. Move Table 1 to appendix.
- For the results section please follow the same order as outlined in the statistics section. Tables 2-6 are very disorganized. Please clean up – two tables should be enough. For instance, why do you report gender twice (table 2 and 6)? In table 6 – better report the range of responses (n) and response rates across nursing homes in a narrative. For table 7 – please also report relative risk ratios since these are easier to interpret. Furthermore, provide a short narrative eluding to the size of the found effects Please move table 8 to the appendix, the estimates are not much
informative.
- The discussion should start with a clear summary of the main results -descriptive and inferential.

Please also see annotations in the comments...

1. Nakagawa S, Schielzeth H. Repeatability for Gaussian and non-Gaussian data: a practical guide for biologists. Biological Reviews. 2010;85(4):935-56.
2. Snijders TAB, and Bosker, Roel J. Multilevel Analysis: An Introduction to Basic and Advanced Multilevel Modeling, second edition. London etc.: Sage Publishers; 2012.
3. Hox JJ, Moerbeek M, Van de Schoot R. Multilevel analysis: Techniques and applications: Routledge; 2010.
4. Bryk AS, Raudenbush SW. Hierarchical linear models: Applications and data analysis methods: Sage Publications, Inc; 1992.

---

## Round 0.4 · accepted · Accept

Dear Dr. Sandvoll,

Thank you for the revised manuscript, which has now addressed all points appropriately. Thank you for your patience in the process.

Best wishes
Michael Simon